# A Systematic Review and Meta-Analysis of the Inhibitory Effects of Plant-Derived Sterilants on Rodent Population Abundance

**DOI:** 10.3390/toxins14070487

**Published:** 2022-07-15

**Authors:** Xuanye Wen, Shuai Yuan, Limei Li, Quanhua Dai, Li Yang, Fan Jiang, Xiao Lin

**Affiliations:** 1Center for Biological Disaster Prevention and Control, National Forestry and Grassland Administration, Shenyang 110031, China; wenxuanye_1116@163.com (X.W.); a202218540290493@163.com (F.J.); 2College of Grassland, Resources and the Environment, Inner Mongolia Agricultural University, 29 Erdos East Street, Saihan District, Hohhot 010010, China; yuanshuai2020@163.com; 3Jilin Provincial Academy of Forestry Science, Changchun 130031, China; lilimei19820406@163.com; 4Shaanxi Province Liuba County State-Owned Zhakou Stone Forestry Field, Hanzhong 723000, China; daiquanhua55@126.com; 5Forestry and Grassland Workstation of Management General Station in Xinjiang Production and Construction Crops, Urumqi 830000, China; li_yang543@163.com

**Keywords:** triptolide, curcumol, capture rate, pregnancy rate, meta-analysis

## Abstract

Owing to their low minimal environmental risk and other ethical considerations, plant-derived sterilants are used to control rodent populations. However, the effects of plant-derived sterilants are not immediate, and their efficacy on rodent control is controversial, which negatively affects sterilant research and application. Here, a meta-analysis of the available literature was conducted to evaluate the effects of two plant-derived sterilants, triptolide and curcumol, on rodent populations. Using a random-effects and a fixed-effects model, we calculated the weighted mean difference (WMD) and relative risk (RR) and their corresponding 95% confidence intervals (95% CIs). After the application of plant-derived sterilants, the rodent population density tended to decrease. Three outcome-related measures in rodents, i.e., capture rate (RR = 0.31, 95% CI [0.20, 0.47]), pregnancy rate (RR = 0.49, 95% CI [0.40, 0.61]), and sperm survival rate (WMD = −17.53, 95% CI [−28.96, −6.06]), significantly decreased, as shown by a significant reduction of ovarian, uterine, and testicular organ coefficients. However, the number of effective rodent holes did not change significantly after the interventions, indicating that the studied sterilants did not directly eradicate the rodent populations. This study provides a theoretical basis for elucidating the inhibitory mechanisms of plant-derived sterilants on rodent populations and for the rational use of these sterilants.

## 1. Introduction

Rodent infestation is a substantial global issue. Widely distributed across the world, rodents cause damage to agriculture, forestry, and livestock production and spread various natural-focal diseases, including plague, leptospirosis, epidemic hemorrhagic fever, and scrub typhus [1]. It is necessary to take measures to control rodent damage in order to protect human health and wealth. However, from the perspective of ecosystems as a whole, rodents are primary consumers in the food chain and prey of many natural enemies, such as small felids, small canines, weasels, snakes, and raptors. Rodents are thus one of the most crucial links in the maintenance of ecosystem energy flows and ecological balance [2,3].

Chemical rodenticides—the most widely used rodent control measure—are widely used throughout the world. Although the application of a chemical bait to regulate rodent damage can limit rodent population abundance, it can also give rise to many other problems. First, satisfactory outcomes are hardly achieved by merely relying on chemical eradication. In certain regions, rodent damage can become more severe, rather than being alleviated. Given the high fecundity of rodents, their rapid eradication can create a ‘vacuum’ in many areas, allowing the remaining rodents to reproduce quickly, restoring or even exceeding the previous population levels [4]. Second, the use of chemical agents raises issues concerning environmental safety and persistence [5]. As anticoagulants are the most-used rodenticides, rodent resistance to anticoagulants has emerged, severely hampering the application of these compounds. Furthermore, the use of anticoagulants to eradicate rodents kills and harms many non-target organisms. Studies have shown that second-generation anticoagulant rodenticides cause strong secondary toxicity to rodents’ natural enemies [6,7,8]. Third, the use of chemical agents brings about prominent ethical problems. Western societies usually regard animals as objects of ethical concern [9], and the use of rodenticides conflicts with animal welfare [10]. The application of chemical agents to eradicate rodents can cause excessive suffering to rodents and is considered an inhumane control measure.

Plant-derived sterilants are rodent control agents that are produced from plants and their extracts. Compared with conventional chemical agents, plant-derived sterilants have obvious advantages, such as being more environmentally friendly and having more persistent control effects [11]. At present, 577 plant species belonging to 122 families are known for their effects on fertility regulation [12]. Among them, triptolide (TR) and curcumol are the most widely used in rodent control. TR is a diterpene epoxide compound with high activity extracted from *Tripterygium wilfordii* Hook of the family Celastraceae [13]. At a low dosage (50 mg/kg), TR caused infertility in half of the tested male rats. At a high dosage (100 mg/kg), TR rendered all tested rats infertile [14]. Curcumol is an important component of the essential oil of *Rhizoma Curcuma*, a traditional Chinese medicine. Curcumol is naturally derived from Zingiberaceae and, structurally, is a guaiane-type sesquiterpenoid [15]. Curcumol has anti-fertility effects on both male and female rats, inducing abnormal ovarian development and prolonged follicular development in female rats and decreasing sperm motility, density, and survival rate and increasing sperm deformity rate in male rats.

Throughout the many years of research on the control effects of animal populations by sterilants, some successful cases of their use to control rodents under specific habitat conditions have been reported. However, in many more cases, the desired outcomes were not achieved due to various factors [16,17]. The purpose of rodent fertility control is to maximize the reduction in their fertility. However, we still do not sufficiently understand the potential of fertility control; therefore, the question of achieving infertility in rodents without killing them is often raised. In addition, due to issues related to the diversity and specificity of rodents and the palatability of sterilants [18,19], more in-depth research on the control effects of sterilants on rodent population abundance is required. A meta-analysis to systematically evaluate the inhibitory effects of two plant-derived sterilants on rodent population abundance can facilitate the sensible use of plant-derived sterilants and provide a scientific basis for comprehensive rodent control and management.

## 2. Results

### 2.1. Literature Search

A total of 2164 relevant studies were obtained from an initial literature search. After multiple levels of screening, 14 studies were included in the analysis. The literature screening process is shown in Figure 1.

### 2.2. Basic Characteristics of the Included Studies

A total of 14 relevant studies were included in this meta-analysis [20,21,22,23,24,25,26,27,28,29,30,31,32,33]. Study information, such as rodent species, study type, bait used, and outcome measures, is provided in Table 1.

### 2.3. Meta-Analysis

#### 2.3.1. Meta-Analysis of the Rodent Capture Rate

Ten studies reported the post-intervention rodent capture rate. An analysis of heterogeneity among the 10 included studies resulted in I^2^ = 17.5% and *p* = 0.292, indicating small between-study heterogeneity, and hence, the fixed-effects model was adopted to generate the combined effect size. We obtained RR = 0.31 (95% confidence interval (CI) ranging from 0.20 to 0.47), indicating that the rodent capture rate was significantly lower in the treatment group than in the control group (Figure 2).

Due to the large number of included studies, publication bias could be assessed by visually inspecting a funnel plot. The funnel plot showed an asymmetrical distribution of the data points, suggesting a small publication bias (Figure 3). Egger’s test was used to evaluate the publication bias, and the result, i.e., *p* = 0.326, indicated no publication bias for the included studies.

#### 2.3.2. Meta-Analysis of the Rodent Pregnancy Rate

Nine studies reported the post-intervention rodent pregnancy rate. An analysis of heterogeneity among the nine included studies resulted in I^2^ = 69.8% and *p* = 0.000, indicating significant between-study heterogeneity, and hence, the random-effects model was adopted to generate the combined effect size. We obtained RR = 0.49 (95% CI ranging from 0.40 to 0.61), indicating that the rodent pregnancy rate was significantly lower in the treatment group than in the control group (Figure 4).

Due to the small number of included studies (fewer than 10 publications), publication bias could not be effectively assessed by visually inspecting the funnel plot. Egger’s test was instead used to evaluate the publication bias; the result, *p* = 0.000, indicated that a publication bias was present and that some publications yet to be published were not included.

#### 2.3.3. Meta-Analysis of the Rodent Sperm Survival Rate

Two studies reported the post-intervention rodent sperm survival rate. An analysis of heterogeneity between the two included studies resulted in I^2^ = 99.7% and *p* = 0.000, indicating significant between-study heterogeneity, and hence, the random-effects model was adopted to generate the combined effect size. The results showed that the weighted mean difference (WMD) was −17.53 (95% CI ranging from −28.96 to −6.06), indicating that the rodent sperm survival rate was significantly lower in the treatment group than in the control group (Figure 5).

#### 2.3.4. Meta-Analysis of the Number of Effective Rodent Holes

Two studies reported the post-intervention number of effective rodent holes. An analysis of heterogeneity between the two included resulted in I^2^ = 99.8% and *p* = 0.000, indicating significant between-study heterogeneity, and hence, the random-effects model was adopted to generate the combined effect size. We obtained WMD = −101.93 (95% CI ranging from −277.91 to −74.06), indicating that the number of effective rodent holes in the treatment group did not significantly differ from that in the control group (Figure 6).

#### 2.3.5. Meta-Analysis of Rodent Organ Coefficients

Meta-analyses were performed for different rodent organ coefficients. The results showed that for the ovarian organ coefficient, WMD = −0.02 (95% CI ranging from −0.03 to −0.01). This result indicates that the rodent ovarian organ coefficient was significantly lower in the treatment group than in the control group. For the uterine organ coefficient, we found WMD = 0.68 (95% CI ranging from −0.75 to 2.11). This result indicates that the rodent uterine organ coefficient in the treatment group did not significantly differ from that in the control group. For the testicular organ coefficient, we obtained WMD = −0.21 (95% CI ranging from −0.24 to −0.18). This result indicates that the rodent testicular organ coefficient was significantly lower in the treatment group than in the control group (Figure 7).

## 3. Discussion

In the 21st century, fertility control has become a humane, sustainable technology with few negative effects for controlling rodent damage and thus has attracted substantial attention. Knipling [34] first proposed making male rodents sexually sterile to control rodent damage. Davis [35] studied the application of chemical sterilants, such as triethylenemelamine, to control the population abundance of *Rattus norvegicus*. Until the 1970s, chemical sterilants were the primary research focus. Subsequently, research on fertility control basically remained stagnant, due to the following reasons. First, most chemical sterilants are poorly palatable and difficult to apply. Second, chemical sterilants generally have toxic side effects, cause environmental pollution, and may harm non-target organisms. Third, many chemical sterilants are only effective for males, and the polygamy of rodents leads to an unsatisfactory control by sterilants. Fourth, chemical sterilization typically requires the artificial breeding of sterilized rodents prior to application, and thus, its economic benefits are low [36,37,38].

In the 1980s, some researchers started to study plant-derived sterilants and examined gossypol and chemicals from the tuber *Trichosanthes kirilowii* as the main plant-derived sterilants, with initial success. Since then, plants including *Achyranthes bidentata*, *Curcuma longa*, *Phyllanthus emblica*, *Andrographis paniculata*, and their extracts have been used in rodent fertility control [39,40,41] but have not been applied on a large scale due to limitations such as high costs and production difficulties. After intense research and development, curcumol and TR were released in 2006 and 2009, respectively. Compared with other plant-derived sterilants, these sterilants have significantly better palatability and allow a sustained control. Results of field tests also demonstrated their excellent anti-fertility effects [42]. However, in practical applications, the promotion and use of plant-derived sterilants are extremely challenging. As sterilants do not directly cause rodent mortality, their effects on rodent population control are not immediate. Therefore, to control rodent populations, people highly prefer chemical baits, which cause more environmental pollution, are less safe, and pose a greater risk of drug resistance than plant-derived sterilants. Hence, a systematic analysis of the efficacy of plant-derived sterilants in controlling damage caused by rodents would help promote the application of non-toxic and harmless measures for rodent damage control and address the social and environmental problems brought about by chemical pesticides.

In this study, a meta-analysis was conducted to evaluate the control of rodent damage by TR and curcumol. We included 14 studies involving 18 rodent species. The results of the meta-analysis indicated that the two sterilants exerted significant effects on both male and female rodents. These effects included a reduction in the testicular organ coefficient and sperm survival rate of males as well as in the ovarian organ coefficient and pregnancy rate of females. Sterilized female rodents could not reproduce, and sterilized male rodents could not reproduce with fertile female rodents, thereby decreasing the overall reproductive rate of the rodent population [43,44]. After feeding on the two types of bait, the number of effective rodent holes for the treatment group did not differ from that for the control group. This indicates that sterile individuals continued to occupy nests, consume resources, maintain tense social pressure, and inhibit population recovery [45]. Sterile female and male rodents could not reproduce, which resulted in a reduced number of rodent offspring and a reduced rodent capture rate in the following season or year, thereby decreasing rodent population abundance.

In terms of controlling rodents, fertility control has greater potential than conventional rodenticides [46,47,48]. In small environments, such as shops, airports, and warehouses, it may be desirable to kill all rodents. However, when considering the entire ecosystem, making rodents sterile can better achieve the purpose of controlling their population abundance [49]. Throughout 2000 years of human practice of rodent eradication, it has appeared that, even when all eradication means are pursued, it is impossible to eliminate all rodents in a certain environment [50,51]. Therefore, the remaining rodents will continue to survive. When their population density declines to a very low level, it is very challenging to further eradicate rodents. In this case, fertility control can play a role in competitive exclusion through reproductive interference [52]. The combined use of rodenticides and sterilants can first eliminate most individuals and render the remaining rodents sterile. Due to competitive reproductive interference by sterile rodents, the few remaining fertile rodents and normal rodent migrants are unable to reproduce. This allows for the long-term control of rodents.

This meta-analysis revealed heterogeneity in the rodent capture rate, pregnancy rate, sperm survival rate, and number of effective rodent holes among the included studies. There can be several reasons for these differences: 1. research methodology was not uniform among studies; 2. the environmental conditions were not consistent among studies; 3. few studies considered the same outcome measures; 4. the inclusion of studies was not comprehensive, and therefore, some data in to-be-published, published, and grey literature were not included in the analysis. In addition, due to different living habits and habitat environments of different rodent species, differences in survey methods might have contributed to the aforementioned heterogeneity in outcome measures among studies.

The control of rodent damage requires an emphasis on economic benefits and, even more, on ecological and social benefits. As research on technologies of rodent fertility control continues, the indirect effects of plant-derived sterilants on the reduction of rodent population density will become more apparent. The current problems associated with fertility control and the environmental problems associated with chemical rodenticide application will be overcome. As an essential measure for the comprehensive control of damage caused by rodents, fertility control appears to have broad prospects in future applications.

## 4. Conclusions

In conclusion, our study provides new evidence of the efficacy of plant-derived sterilants against rodents, demonstrating that triptolide and curcumol can effectively disrupt the reproductive system of rodents without directly killing them. Combined with the significant advantages of plant-derived sterilants in terms of environmental friendliness, more consideration should be given to their application in nature reserves, pastures, or areas where large-scale rodent infestations are predicted to occur.

## 5. Material and Methods

### 5.1. Registration

This meta-analysis was designed based on the PRISMA statement [53] and registered with the International Prospective Register of Systematic Reviews (PROSPERO, https://www.crd.york.ac.uk/PROSPERO, accessed on 20 January 2021).

### 5.2. Inclusion and Exclusion Criteria

In this meta-analysis, randomized controlled trials were the only study type selected, with rodents as the research objects. Intervention measures refer to the application of curcumol or TR baits, and conventional baits were employed in the control group. Outcome measures were rodent capture rate, pregnancy rate, number of effective holes, ovarian organ coefficient, sperm survival rate, uterine organ coefficient, and testicular organ coefficient.

The inclusion criteria were as follows:

(1) Studies on the effects of curcumol or TR baits on rodent reproduction; and (2) randomized controlled trials.

The exclusion criteria were as follows:

(1) Studies in which the research objects were not clear and could not be ascertained to be rodents; (2) outcome measures selected for this meta-analysis were absent; (3) data could not be extracted, or the full text was not available; (4) duplicate publications; and (5) reviews and abstracts.

### 5.3. Search Strategies

Using “rodentia”, “rodents”, “rat”, “mouse”, “mice”, “curcumol”, and “triptolide” as keywords, Chinese databases, including China National Knowledge Infrastructure (CNKI), Wanfang Data, and the VIP database, were searched. English databases searched included PubMed, Web of Science, ProQuest, and SpringerLink. The timeframe of the search was from the inception of the databases to December 2020. Search languages were limited to Chinese and English.

### 5.4. Literature Screening and Data Extraction

First, based on the inclusion and exclusion criteria, 2 researchers independently screened the retrieved literature, which was then cross-checked. Differing opinions on the same studies were evaluated by a third party and were further discussed to reach an agreement. Two researchers separately extracted relevant information from the included literature, including author names, year of publication, region, study design, intervention measures, sample size, and outcome measures.

### 5.5. Statistical Analysis

A meta-analysis of the data was conducted using Stata 16.0. Heterogeneity among studies was assessed by the χ^2^ test combined with the I^2^ statistic. When *p* > 0.1 and I^2^ < 50%, between-study heterogeneity was acceptable, and the meta-analysis was performed using the fixed-effects model. If *p* < 0.1 and I^2^ > 50%, between-study heterogeneity was high, and the meta-analysis was performed using the random-effects model. Publication bias of the included studies was assessed by Egger’s test. *p* > 0.05 indicated that publication bias was not significant.

## Figures and Tables

**Figure 1 toxins-14-00487-f001:**
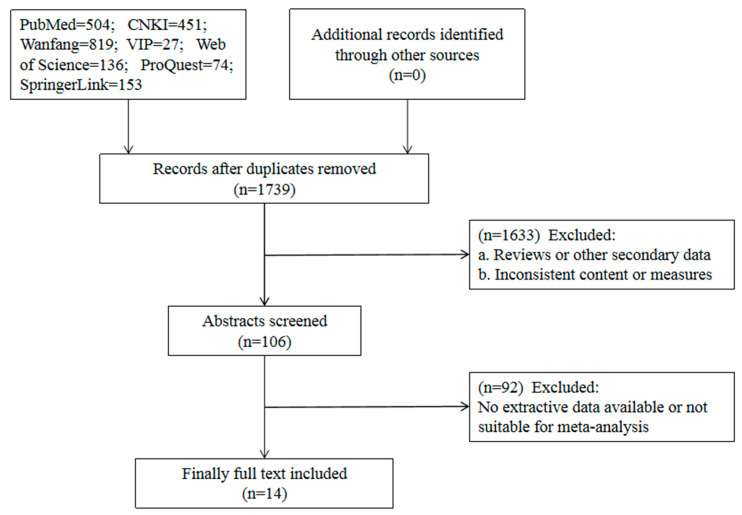
Flow chart of the literature screening process.

**Figure 2 toxins-14-00487-f002:**
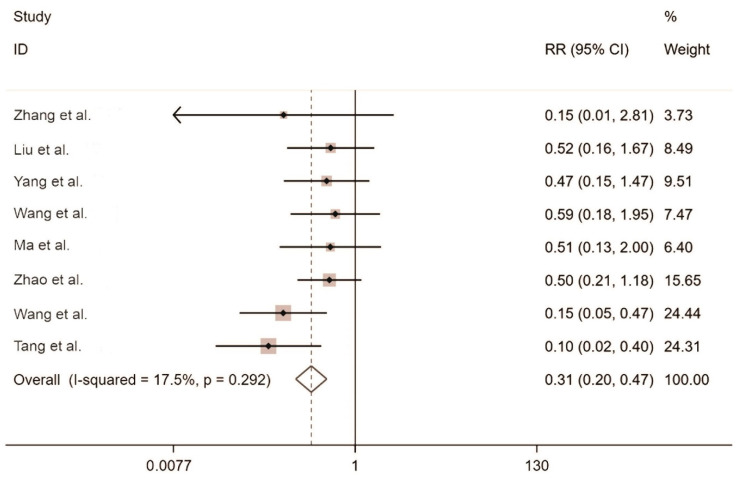
Forest plot of the rodent capture rate. Zhang et al. [32], Liu et al. [25], Yang et al. [31], Wang et al. [29], Ma et al. [26], Zhao et al. [33], Wang et al. [30], Tang et al. [27] The black dots are the odds ratio, the squares are the weight sizes, the hollow diamond indicates the overall effect size, and the red dashed line indicates the average of the overall effect size.

**Figure 3 toxins-14-00487-f003:**
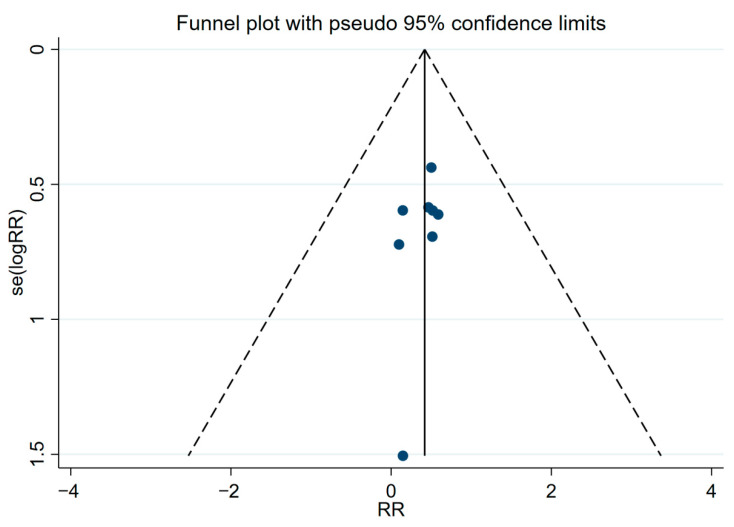
Funnel plot for the rodent capture rate.

**Figure 4 toxins-14-00487-f004:**
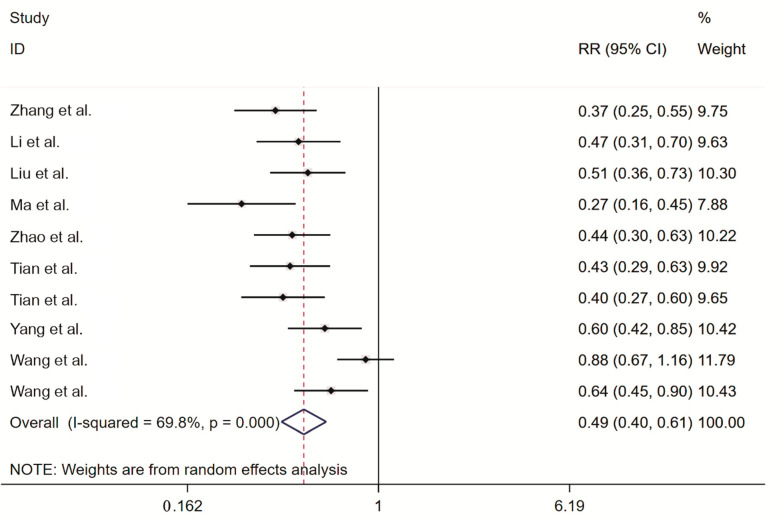
Forest plot of the rodent pregnancy rate. The intervention measure in Tian et al. [28]a was a curcumol bait; the intervention measure in Tian et al. [28]b was a triptolide bait. Zhang et al. [32], Li et al. [32], Liu et al. [25], Ma et al. [26], Zhao et al. [33], Tian et al. [28]a, Tian et al. [28]b, Yang et al. [31], Wang et al. [29], Wang et al. [30]; The black dots are the odds ratio, the squares are the weight sizes, the hollow diamond indicates the overall effect size, and the red dashed line indicates the average of the overall effect size.

**Figure 5 toxins-14-00487-f005:**
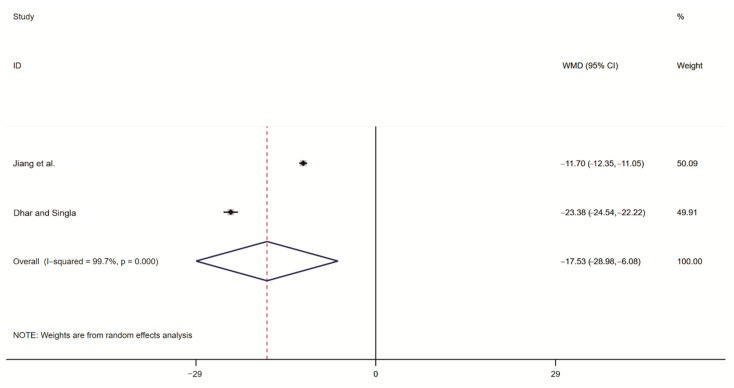
Forest plot of the rodent sperm survival rate. Jiang et al. [22], Dhar and Singla [20], The black dots are the odds ratio, the squares are the weight sizes, the hollow diamond indicates the overall effect size, and the red dashed line indicates the average of the overall effect size.

**Figure 6 toxins-14-00487-f006:**
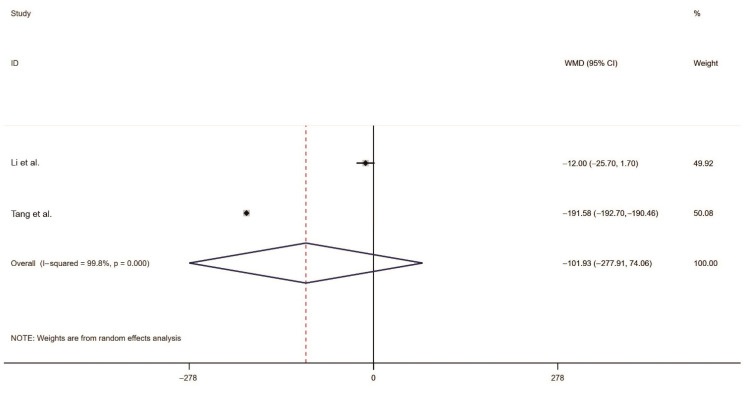
Forest plot of the number of effective rodent holes. Li et al. [24], Tang et al. [27] The black dots are the odds ratio, the squares are the weight sizes, the hollow diamond indicates the overall effect size, and the red dashed line indicates the average of the overall effect size.

**Figure 7 toxins-14-00487-f007:**
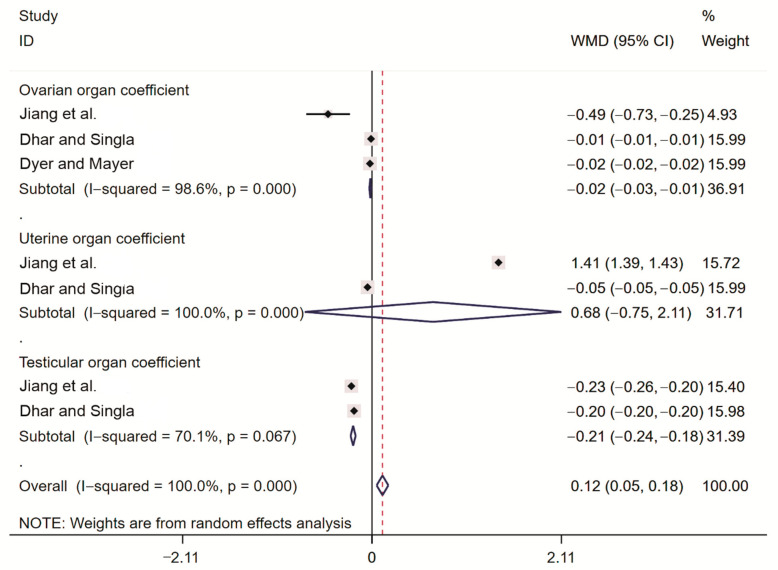
Forest plot of the rodent organ coefficients. Jiang et al. [22], Dhar and Singla [20], Dyer and Mayer [21], The black dots are the odds ratio, the squares are the weight sizes, the hollow diamond indicates the overall effect size, and the red dashed line indicates the average of the overall effect size.

**Table 1 toxins-14-00487-t001:** Basic characteristics of the included studies.

Study	Country	Study Type	Intervention Measure	Species	Outcome Measures
Zhang et al. [32]	China	randomized controlled trial	curcumol bait	*Apodemus draco* *Eothenomys miletus* *Rhombomys opimus*	capture rate pregnancy rate
Li et al. [23]	China	randomized controlled trial	curcumol bait	*Apodemus draco* *Eothenomys miletus*	pregnancy rate number of effective holes
Liu et al. [25]	China	randomized controlled trial	curcumol bait	*Apodemus peninsulae*	capture rate pregnancy rate
Li et al. [24]	China	randomized controlled trial	curcumol bait	*Ochotona curzoniae* *Cricetulus longicaudatus*	number of effective holes
Yang et al. [31]	China	randomized controlled trial	triptolide bait	*Cricetulus longicaudatus* *Apodemus peninsulae* *Niviventer niviventer* *Apodemus agrarius*	capture rate pregnancy rate
Wang et al. [29]	China	randomized controlled trial	curcumol bait	*Cricetulus longicaudatus*	capture rate pregnancy rate
Ma et al. [26]	China	randomized controlled trial	curcumol bait	*Tscherskia tritonde* *Apodemus agrarius*	capture rate pregnancy rate
Zhao et al. [33]	China	randomized controlled trial	curcumol bait	*Ochotona curzoniae*	capture rate pregnancy rate
Tian et al. [28]	China	randomized controlled trial	curcumol bait, triptolide bait	*Spermophilus dauricus*	capture rate pregnancy rate
Jiang et al. [22]	China	randomized controlled trial	curcumol bait	*Lasiopodomys brandtii*	ovarian organ coefficient uterine organ coefficient testicular organ coefficient sperm survival rate
Wang et al. [30]	China	randomized controlled trial	triptolide bait	*Rattus losea* *Rattus norvegicus* *Rattus flavipectus*	capture rate pregnancy rate
Dhar and Singla [20]	India	randomized controlled trial	triptolide bait	*Bandicota bengalensis*	ovarian organ coefficient uterine organ coefficient testicular organ coefficient sperm survival rate
Dyer and Mayer [21]	USA	randomized controlled trial	triptolide bait	*Rattus norvegicus*	ovarian organ coefficient
Tang et al. [27]	China	randomized controlled trial	curcumol bait	*Pitymys leucurus*	capture rate number of effective holes

## Data Availability

Not applicable.

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
