# Peer review of "A Systematic Review and Meta-Analysis of the Inhibitory Effects of Plant-Derived Sterilants on Rodent Population Abundance"

_toxins, 2022, doi:10.3390/toxins14070487_

Round 1

Reviewer 1 Report

„Inhibitory Effects of Plant-derived Sterilants on Rodent Population Abundance: A Systematic Review and Meta-analysis“

Is more of a statistical review article, so this should be emphasized in the title in the first part. When the title is written this way, the concept of the review article becomes much clearer. Perhaps the title should read something like this: A meta-analysis of articles on "Inhibitory effects of plant sterilizers on rodent population numbers".
It would be good to write a Conclusion to add transparency to the article. All other parts of the article are written correctly.

Author Response

Thank you for your valuable suggestions. According to your suggestion, we have revised the title of the paper. In the last part of the paper, we add the conclusion to ensure that readers can more clearly understand the specific results of this paper.

Reviewer 2 Report

In the present review, a meta-analysis was conducted to evaluate the control of rodent damage by TR and curcumol. Only 14 studies were included, and this is a relevant limitation. Are the authors sure of this limited number of articles to be considered? Another section to be remodulated is the references. They seem so old. I suggest the authors to improve the quality of the review by implementing other more recent studies.

Author Response

Thank you very much for your patience in reviewing this paper. You have made a great contribution to the improvement of this paper. Please allow me to express my heartfelt gratitude to you.

1) At present, meta-analysis needs to comply with strict process regulations. This meta-analysis was designed based on the PRISMA statement and registered with the PROSPERO.According to the requirements of this rule, we can only select the literature that meets the preset conditions to be included in the study. In addition, the methods and results adopted by a large number of existing studies do not meet the basic requirements of meta-analysis, so they are not included in this study. Compared with other similar types of analysis, 14 studies of medium quantity, which can meet the requirements of clear research results.

Other studies based on PRISMA are as follows::

Including 11 studies: Heneghan C, et al.Self-monitoring of oral anticoagulation: systematic review and meta-analysis of individual patient data. The Lancet, 2012, 379(9813):3 22-334.

Including 16 studies: D'Silva KM, et al. Proton pump inhibitor use and risk for recurrent Clostridioides difficile infection: A systematic review & meta-analysis. Clinical Microbiology and Infection, 2021, 27(5): 697-703.

Including 12 studies: Shim M-S, Noh D. Effects of Physical Activity Interventions on Health Outcomes among Older Adults Living with HIV: A Systematic Review and Meta-Analysis. International Journal of Environmental Research and Public Health. 2022; 19(14):8439.

Including 10 studies: Hung K-C, et al. Association between Fibrinogen-to-Albumin Ratio and Prognosis of Hospitalized Patients with COVID-19: A Systematic Review and Meta-Analysis. Diagnostics. 2022; 12(7):1678.

2)As Reviewer suggested,We read references carefully and updated according to the content of the paper. In particular, the references of the 1960s and 1970s in the paper has been updated.

Reviewer 3 Report

The article under review focuses on an impactful research area, the control of rodents through plant-based compounds that decrease fertility.

The title is appropriate for the paper. The introduction offers brief and up-to-date information on the subject at hand. Material and methods are standard and well described.

Results are  presented in a clear fashion, using state-of-the-art statistical representations that are customary in meta-analyses. Discussions include obtained results in the international context.

Conclusions are supported by the presented data, that are meaningful for the long-term control of rodents.

Author Response

Thank you very much, and on behalf of all the authors of this paper, I would like to express my sincere thanks for your careful review.